# Toward improvement of knowledge of financial conflicts of interest in a large medical school in France

J. F. Alexandra[1,2], D. Roux[3,4]☯, H. Maisonneuve[5]☯, B. Chousterman[2,5], P. Ruszniewski[6], D. Dreyfuss[2,3,7] *

1 APHP, Hôpital Bichat Claude Bernard, Service de Médecine Interne DMU Victoire Paris, Paris, France, 2 Université Paris-Cité, Paris, France, 3 AP-HP, Hôpital Louis Mourier, DMU ESPRIT, Service de Médecine Intensive Réanimation, Colombes, France, 4 Université Paris Cité, INSERM UMR-S1151, CNRS UMR-S8253, Institut Necker Enfants Malades, Paris, France, 5 Hervé Maisonneuve, MD, Scientific Editor & Consultant, Paris, France, 6 Doyen de l'UFR de Médecine, Université Paris Cité, Pairs, France, 7 French National Institute of Health and Medical Research (INSERM), UMR_S1155, Common and Rare Kidney Diseases (CORAKID), Hôpital Tenon, Sorbonne Université, Paris, France

☯ These authors contributed equally to this work.
* didier.dreyfuss@aphp.fr

**Data Availability Statement:** All relevant data are within the manuscript and its Supporting Information files.

## Abstract

### Introduction

Conflict of interests (COIs) adversely affect the integrity of science and public health. The role of medical schools in the teaching and management of COIs has been highlighted by the publication of an annual evaluation of American medical schools based on their COIs policies by the American Medical Student Association (AMSA). A deontological charter was adopted by French medical schools in 2018 but its impact on COI comprehension by students and its effects on COI prevention were not evaluated.

### Methods

A 10-item direct survey was conducted among about 1000 students in Paris-Cité University in order to investigate the respect of the charter regarding COIs both in the medical school and in affiliated teaching hospitals.

### Results

Cumulative results show a satisfying respect of prevention policies regarding COIs in the medical school and hospitals despite the fact that the existence of the charter and its major aspects were insufficiently known. Disclosure of COIs by teachers was insufficient.

### Conclusion

This first direct study among students shows better results than expected according to current non-academic surveys. Moreover, this study demonstrates the feasibility of this kind of survey whose repetition should be an appropriate tool to improve the implementation of the

**Funding:** the author(s) received no specific funding for this work.

**Competing interests:** the authors have declared that no competing interests exist.

charter within medical schools and teaching hospitals, in particular mandatory disclosure of COIs by teachers.

## Introduction

The impact of conflict of interests (COIs) on public health has become a major concern over the past fifteen years [1–3]. Starting from 2007, the American Medical Student Association (AMSA) published an annual evaluation of American medical schools based on their COIs policies. Such initiative induced major shifts in policies in the recent years [4]. In contrast with the strong policies of COIs management in many North American medical schools, a recent review showed that COIs prevention policies are ranked poorly [5] in European medical schools or teaching hospitals. In Belgium, for instance, a recent study reported little transparency regarding potential COIs and only isolated initiatives to protect student from pharmaceutical promotion [6].

Research works have highlighted the deleterious effects of conflicts of interest on practice, research and training in medicine in France [7]. A web-based study conducted in 2012 [8] reported that medical students (preclinical, clinical and residents) are insufficiently aware of potential bias that COIs may pose with respect to drug prescriptions and want to be informed about the COIs of their lecturers.

A 2017 national survey of French medical schools based on AMSA criteria [9] was conducted by a non-academic association named FORMINDEP (which stands for FORmation Medicale INDEPendante: Independent Medical Training). This association aims at both promotion of evidence-based medicine and prevention of the influence of industry's economic interests on patients, physicians, medical students and public policy makers [10]. The same year, the conference of Deans of Schools of Medicine and Dentistry (a national institution that brings together the Deans of the 34 French medical schools and of the 16 Schools of Dentistry) [11] adopted an ethical charter. This charter meets an ethical requirement, particularly with regard to scientific and professional integrity and to COI avoidance and promotes the independence of medical training through an institutional and academic framework. A summary of the charter is available in S1 Appendix.

Formindep survey was reiterated in 2018 [12] and 2021 [13]. In each instance, the conclusions were sobering as they indicated a quasi-complete lack of academic policy to prevent COIs, in particular those related to drugs and devices despite previous adoption of the charter. Moreover, we had no information on medical student's awareness of the charter.

In order to promote the knowledge and application of the charter, several measures have been taken by the university since 2017:

1. A 2-hour mandatory course dealing with the relationships with industry and with the management and prevention of COIs was organized for all 4[th] year students, starting in 2018 and given by one of us (DD). Definition and examples of financial COIs and the way to disclose and avoid it are presented. Available literature on the influence of COIs on scientific integrity is discussed. The main aspects of the charter are detailed and the link to the charter and its summary is recalled.

2. The charter was sent to all academics of medical school of Paris Cité University in June 2019. This university, one of the largest in France has more than 25 000 students, 1400 teachers and 34 affiliated hospitals or research centers. They were asked to read and sign it. In addition, for the sake of simplicity, a summary of the charter was drafted by the

Commission of Ethics of our Medical School. This summary underlined major points of the charter and in particular the prevention and management of COIs. It was also sent to all academics. In addition, both the complete charter and its summary were posted on our Medical School website (available at . . .). The administration of the medical school and the Heads of all medical departments of the University-affiliated hospitals were requested to display the summary of the charter in all premises where teaching was provided.

3. Every year from 2019, all newly recruited assistant professors attend a session of formation. The course is mainly devoted to explain pedagogic tools and policy of our Medical School. It is also the occasion to sensitize young teachers to the importance of COI understanding and management during their teaching activity as well as for their own behaviour.

Four years after the adoption of these measures, we wanted to evaluate their effects. We subsequently conducted a survey of medical students in their 4th and 5th year of medical cursus in our medical school which is affiliated to Université Paris Cité between December 2021 and January 2022. The aim of this survey was to evaluate student awareness of the charter and to what extent it was applied. In particular, we aimed at investigating how COIs prevention policies were implemented in both university premises and teaching hospitals of our medical school

## Methods

1. The present survey was prepared in 2021. The Commission on Ethics of our medical school developed a 10-item questionnaire with two objectives: evaluation of student knowledge of the charter and evaluation of the frequency of its display in all teaching facilities (including medical school premises and university-affiliated teaching hospitals. The questionnaire aimed at evaluating every point of the charter that deals with the problem of COIs. To allow a nuanced analysis of the questionnaire all questions were on a 4-level graduation scale from "absolutely yes" to "absolutely not" (the last question only could be answered by yes or no)

2. All students in their 4th and 5th year of graduation received an e-mail informing them that they were asked to participate in an anonymized survey of the charter at the time of their examinations. They were free to answer or not.

3. At the time of the digital exams, within the university premises, students were asked to complete the questionnaire. Answers were anonymized for analysis.

4. Answers where collected by the software of the university and subsequently analysed without the need for any specific statistical methodology.

Patient and public involvement: No patient involved in our study.

French law on research involving human subjects applies only to research on patients and to surveys on health data. It therefore does not apply to an anonymized survey of medical student feeling about their medical school policy. No personal data was involved. In a study like ours, no IRB approval is required by French law. Obviously students were informed of the purpose of the survey and were perfectly free to refuse it.

## Results

A total of 710 out of 1000 registered students (71.0%) in their 4[th] year of graduation provided at least one answer. A total of 552out of 852 (64.7%) in their 5[th] year of graduation provided at least one answer. Overall, there were between 1179 and 1186 answers per question for 4[th] and

**Table 1.**

| | *Absolutely yes* | *Rather yes* | *Rather not* | *Absolutely not* |
|---|---|---|---|---|
| *Do your teachers indicate their COIs before the course* | 246 (21%) | 445 (37%) | 342 (29%) | 149 (13%) |
| *Do the courses given to you name the molecules in INN or in therapeutic class?* | 274 (23%) | 730 (63%) | 143 (12%) | 21 (2%) |
| *Do your teaching materials (books, handouts) from university contain advertising for the drug industries?* | 18 (2%) | 43 (4%) | 472 (40%) | 644 (54%) |
| *Are you aware of the university's Charter of ethics?* | 172 (15%) | 353 (30%) | 353 (30%) | 305 (25%) |
| *Is the ethics charter clearly posted on university premises?* | 62 (5%) | 245 (21%) | 607 (52%) | 251 (22%) |
| *Is the ethics charter clearly posted on the premises of your teaching hospital?* | 51 (4.5%) | 156 (13.5%) | 515 (44%) | 447 (38%) |
| *Do you have meetings with drug manufacturers within your teaching hospital?* | 137 (12%) | 301 (25%) | 344 (29%) | 401 (34%) |
| *Do you benefit from services in kind (breakfast, buffet, gifts, etc.) from manufacturers within your teaching hospital?* | 44 (4%) | 102 (9%) | 229 (19%) | 807 (68%) |
| *Do you benefit from services in kind (breakfast, buffet, gifts, etc.) from manufacturers within the university premises?* | 0 | 33 (3%) | 192 (16%) | 939 (81%) |
| | Yes | | No | |
| *Do you know the procedure for reporting cases of breaches of the principles of the charter in your medical school (discrimination, harassment, conflict of interest)?* | 187 (16%) | | 994 (84%) | |

5<sup>th</sup> year students, respectively (63,7% response rate). The number of questionnaires entirely filled was 1179 (64%). Results are shown in **Table 1**.

Eighty four (84%) percent of students indicated that their books and handbooks provided by the medical school teachers denominated products by therapeutic class or International Non proprietary Names (INN). Moreover, 94% declare that these teaching materials (books, handouts) did not contain any advertising. However, students were only 58% to report that teachers declare their COIs when they begin their course.

Ninety percent (90%) declared receiving no gift nor benefit of any kind although 37% still had contact with drug manufacturers in the hospital units. The charter was poorly known and not widely displayed (63% of students declared they did not know it and between 71% and 80% did not know where it was posted in medical school premises nor in their teaching hospitals). Last, 80% of students did not know the procedure for reporting breaches of the charter.

## Discussion

Our study, the first taking directly in account students' perception in France, revealed contrasting but encouraging elements: the spirit of the charter is widely respected although the letter is somewhat poorly known in our university and related teaching hospitals.

The results of the first survey conducted by FORMINDEP [10] were published in 2017 under a self-explicative title: "Conflict of Interest Policies at French Medical Schools: Starting from the bottom". The methodology allowed only indirect evaluation of schools' performance on COI policy: some student organizations reported their appreciation of the implementation of the charter in each of the medical schools via a questionnaire. The Conference of Deans did not answer to this questionnaire despite solicitation [14]. At this time, 9/37 only of French medical schools had either introduced a related curriculum or implemented a COIs-related policy. Of these, only 1 had restrictive policies for any category. FORMINDEP conducted 2

other studies in 2018 and 2021, in the aftermath of their initial survey in order to evaluate the national implementation of the charter [12, 13]. Results were still considered unsatisfactory: of the 36 Medical Schools, only one obtained a score above the average with 18 points out of a possible 34 (based on a 17 criteria scale). Next came 11 schools with a highest score ranging between 10 and 14 points for the highest scores. Similar findings were reported in an independent study conducted in the 32 French teaching hospitals in 2017 [15]. Only 17 had rules and regulations in link with a limited number of items of the charter. Four of them considered implementing a policy and only 2 actually started the implementation. Fifteen had no evidence of COIs policies.

The accuracy of results of previous studies is beyond doubt but may have been affected by some kind of bias: first, as mentioned above, only partial or complete lack of responses was observed for 6 out of 17 criteria. Second, the survey consisted in questionnaires sent to Deans only and on observational data gathered by student organizations. There was no individual student survey. In addition, FORMINDEP studies mainly focused on declarative items, regardless of the actual content of the teachings within university or within hospitals. As a result, some items may have been underestimated. Nevertheless, this must not obscure the fact that teaching of COIs is seldom in French Medical Schools and that the vast majority of students is not aware of the charter. Thus, the persistence of poor results observed in the most recent FORMINDEP survey are unlikely to be explained by methodological bias only.

Several measures were taken over recent years in order to better manage financial COIs. First, a Sales Visit charter was signed in 2014 by the health minister and the drug industry union [16]. Second, a transparency digital register directly inspired by the 2011 sunshine act was implemented in 2018 [17]. This register collects every amounts of money received by doctors from industry whatever the reason (consultancy, travel aid, meals and research contracts).Third, as already alluded to, the Deans' charter explicitly requests that teachers display their COIs at the beginning of every lecture and prohibits the industry representatives from meeting with students, either on the teaching premises or on hospital placement sites. Last, the internal rules of our teaching hospitals (Assistance-Publique-Hôpitaux de Paris) have been amended to be consistent with the provisions of the charter concerning the prohibition of contact between industry representatives and students. This amendment occurred after our survey was conducted.

Our survey, the first one among students in France (contrarily to other countries [18]) was conducted 3 years after implementation of these measures and focused on items reflecting the application and knowledge of the charter based directly on student statements. We chose to focus on 4th and 5th year students since these years are both those where teaching of drugs and devices is predominant and where the students are on internship at the hospital.

Three points deserve mention.

First, the rather good response rate shows the feasibility of this kind of survey. Thus, repetitive studies may be an appropriate tool to measure the implementation of the charter. Second, our results confirm in part those of previous FORMINDEP surveys but do not sustain their poor rating. In almost 90% of courses, drugs are denominated by therapeutic class and/or INN. Students hardly ever receive gifts or benefits in kind. Last, pedagogic tools provided by our medical school are free from advertising. However, a still noticeable number (about one third) of students meet drug sale representatives in teaching hospitals. Upon our request, the Assistance Publique-Hôpitaux de Paris (the hospital network that includes all academic hospitals in the Paris area) has modified its own rule in order to align it with the charter. More recently, French Health regulator (HAS) published guidelines to prevent COI within medical school [19]

We can therefore reasonably expect that meetings of students with sale representative will decrease. On the other hand, poor knowledge of the existence of the charter is prevalent as

more than half students are not aware of its existence (despite the fact that all were supposed to receive it via their e-mail address). A further effort should be made in order to increase the display of the charter in our teaching hospitals and in the premises of our Medical School. Similar efforts should be made, to increase awareness of procedures to declare the cases of charter breach and the obligation to disclose the COIs before the course. Hence, we can assume than despite a poor knowledge of the charter itself, its principles are at least partly respected in our Medical School and its affiliated hospitals.

There are potential pitfalls in our study. First, our study focused only on students' perception of the charter implementation. Thus we did not take the potential influence of partnership between teachers and industry (personal honoraria for counselling or presentations during congresses and sponsoring for research) into account. These ties may affect the integrity of medical education [20] and declaring COIs may not be sufficient to guarantee the independence of professionals [21]. Indeed, teachers are important role models for students, exerting influence on their future prescribing habits and on their future relationship with the pharmaceutical industry [22]. Although transparency is not sufficient to guarantee independence, it is a first mandatory step. Moreover, it has been shown that students in medical schools with strict COIs policies are less likely to experience influence of drug companies [23]. Last, we did not investigate the behavior of the teachers, but they probably have a part of responsibility for the poor results observed in our study. Some teachers, perhaps the seniors, have been accustomed to extensive relationships with industry. They have had less awareness of COIs and have little motivation to change a behavior they have engaged in during their careers. An effort should be made in order to increase awareness of senior academics and promote their observance of the charter. This study seems to be both feasible and reliable and regular repetition could be an adequate tool to measure the evolution of practices. Based on these first results, the areas for improvement are well identified and will focus primarily on knowledge and display of the charter. We are undertaking an information campaign to raise awareness of the charter and will evaluate the results of this campaign in the next academic year. The next analysis will allow us to evaluate the achievement of these objectives.

At the same time, an important point would be to investigate how students are sensitised to COI by their teachers. This is the purpose of a future study.

We assume in accordance with an abundant literature that early interventions such as those implemented in our medical school will foster improved student knowledge of this issue [22–25].

## Conclusions and perspectives

Our study aimed at reinforcing the effects of studies carried out by the FORMINDEP and of recently adopted laws on COIs management. Indeed, a direct and standardized measurement of ethical and deontological charter application may improve its implementation. This study provides clear evidence of feasibility of a wide survey among a large population of medical students. Moreover, the results show that policies preventing COIs are better applied than expected despite imperfect knowledge of the charter. To improve this knowledge, we plan to reiterate the survey in our institution. In addition, we aim at implementing this survey at the national level in all French Medical Schools, under the auspices of the Deans Conference.

We believe that promotion of an official national academic policy is an attainable goal.

## Supporting information

**S1 Appendix. Summary of the charter.**
(DOCX)

## Author Contributions

**Conceptualization:** J. F. Alexandra, D. Roux, B. Chousterman, D. Dreyfuss.

**Formal analysis:** J. F. Alexandra, D. Roux.

**Investigation:** J. F. Alexandra, D. Dreyfuss.

**Methodology:** J. F. Alexandra, D. Roux, D. Dreyfuss.

**Supervision:** H. Maisonneuve, P. Ruszniewski.

**Validation:** P. Ruszniewski.

**Writing – original draft:** J. F. Alexandra, D. Dreyfuss.

**Writing – review & editing:** H. Maisonneuve, D. Dreyfuss.

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
