## [Decision Letter · Decision Letter 0]

31 Jan 2023

PONE-D-22-34803Toward improvement of knowledge of financial conflicts of interest

in a large medical school in FrancePLOS ONE

Dear Dr. alexandra,

Thank you for submitting your manuscript to PLOS ONE. After careful consideration, we feel that it has merit but requires some clarifications and revisions to fully meet PLOS ONE’s publication criteria as it currently stands. Therefore, we invite you to submit a revised version of the manuscript that addresses the points raised during the review process.

ACADEMIC EDITOR: Both reviewers’ comments are helpful to improve the quality of the article. In particular, they indicate some issues that require further clarification, with a special attention on their comments regarding the methods section. In addition, I recommend that you thoroughly review the manuscript for typos and errors. It is often useful to have the text proofread by someone else who can see it with new eyes.

We look forward to receiving your revised manuscript.

Kind regards,

Alberto Molina Pérez, Ph.D.

Academic Editor

PLOS ONE

Journal Requirements:

2. You indicated that ethical approval was not necessary for your study. We understand that the framework for ethical oversight requirements for studies of this type may differ depending on the setting and we would appreciate some further clarification regarding your research. Could you please provide further details on why your study is exempt from the need for approval and confirmation from your institutional review board or research ethics committee (e.g., in the form of a letter or email correspondence) that ethics review was not necessary for this study? Please include a copy of the correspondence as an ""Other"" file.

Reviewers' comments:

Reviewer's Responses to Questions

**Comments to the Author**

1. Is the manuscript technically sound, and do the data support the conclusions?

Reviewer #1: Yes

Reviewer #2: Yes

2. Has the statistical analysis been performed appropriately and rigorously? 

Reviewer #1: I Don't Know

Reviewer #2: Yes

3. Have the authors made all data underlying the findings in their manuscript fully available?

Reviewer #1: Yes

Reviewer #2: Yes

4. Is the manuscript presented in an intelligible fashion and written in standard English?

Reviewer #1: Yes

Reviewer #2: Yes

5. Review Comments to the Author

Reviewer #1: Thank you for the opportunity to review this paper. I enjoyed reading it and it reports on a useful and important study. I have made some suggestions for how its clarity could be improved. Most importantly, there is confusion about what the study actually entailed.

As an initial comment, Charte is Charter (rather than chart)

In the introduction you say there is a quasi-complete lack of academic policy, but in the next sentence you introduce the Charter. This seems contradictory. Do you mean that there is policy that is not implemented? Or that despite agreeing to the Charter no policy actions have been taken? This needs a bit of clarity.

The methods section is confusing. Were items 1-4 part of the study or would they have happened anyway as part of professional training? Here they seem to be presented as part of the experiment but up to this point the paper doesn’t suggest this. My reading of the paper is that the survey of the students is the study, to see whether training their instructors on the Charter led to change in practice or had any sort of trickle down effect to students. If this is the case, the info from 1-4 could go into the introduction. I agree that it is important information but it is not a study method. If the activities in 1-4 were done for the specific purpose of this study then that needs to be made much clearer up front and ideally methodologically justified. Would you have expected that working with instructors would sensitise students to COI? Were instructors asked to train students on COI?

Per number 5, was it compulsory? The response rate in the results suggests not but the wording in 5 suggests yes. Perhaps reword 5 in the methods.

Per number 6, it’s unclear why there are 4 responses to the questions because they are all essentially y/n questions and you subsequently report the results as y/n in each category. Can you talk about why you set up the questionnaire like this?

How did you decide the questions? Are the questions relevant to the Charter? Do they test particular aspects of the Charter?

Discussion:

The second paragraph repeats the information in the introduction and should be removed. In this paragraph you use ref 8 to note that students are unaware of the harms that COI might pose. This would be useful info to have in the introduction (where you use ref 8 to make a very different point).

In para 3 of the discussion you need to include more information about FORMINDEP if you’re going to talk about survey response. Perhaps at the beginning of this para you could say that the survey measures schools’ performance on COI policy. As it was I wasn’t sure what the numbers referred to and it needed some reading between the lines.

P8 “The lack of knowledge of these issues…” – which issues? Is it that they lack knowledge about COI? About the importance of policies mitigating against COI? or don’t care or haven’t made it a priority or something else?

Also p8 you make it sound like items 1-4 in the methods were in fact part of the study? Again, confusing.

P8 is the first place that you mention Charter expectations (eg that teachers should declare at the beginning of each class), though you say you alluded to it. I note in the text that you included the Charter as an appendix but it did not show up in my review. If it is short enough it would be great to include it (or a summary) in a box in the manuscript. It is unlikely that most readers will take the trouble to find an appendix before reading the paper and the paper makes less sense without knowing the contents or even the gist of the Charter.

The pitfall section: The sentence about not taking the potential influence of partnership between teachers and industry into account is confusing. Into account how? Perhaps you mean that you only studied the Charter and its implementation and not COI that are not covered in the Charter? Some clarity here would be beneficial.

Reviewer #2: Thank you for conducting this study, the results of which seem very important to me. I just have a few minor suggestions to further improve the quality of the paper. I will make my recommendations by mentioning systematically and in order the number of the page to which I refer.

Page 3:

"This association aims at both to promotion of evidence-based medicine and prevention of the influence of economic interests of industry on patients, doctors, medical students, and public decision-makers" -> The sentence does not sound good. It could be something like : « This association aims both to promote evidence-based medicine and to prevent the influence of industry's economic interests on patients, physicians, medical students and public policy makers. »

Page 4:

You refer to an appendix but I have not been able to find it.

Methods, point 3: "Implementation of the Chart" -> It is strange to have a specific title for this point and not for the others, it should be harmonized

Page 5:

Methods: In my opinion, only points 5 and 6 really constitute the methodology of the survey presented in the study. The previous points seem to me to be more contextualizing elements. Also, it would be interesting to know a bit more about the creation of the questionnaire: how the 10 items were decided, when exactly the questionnaire was submitted, how did you analyze the results, etc.

Page 7:

Discussion: "the first taking directly in account students’" -> "I would add "on this specific subject" or "in France" because there are many international studies that directly take into account the opinions of medical students on their interactions with industry, the issue of independence and their perceptions. For example: " ext-link-type="uri" xlink:type="simple">https://doi.org/10.1186/s12909-018-1394-9"

Page 9:

First, the rather good response rate shows the feasibility of this survey. - « of this kind of survey » ?

-----

As you can see, my main point is about the presentation of the survey methodology. This section could be reworked in order to better present the different steps that led to the creation of the study.

For the rest, it is a very good paper on a fundamentally important topic.

6. PLOS authors have the option to publish the peer review history of their article (what does this mean?). If published, this will include your full peer review and any attached files.

Reviewer #1: No

Reviewer #2: **Yes: **Lucas Bechoux

---

## [Author Response · Author response to Decision Letter 0]

23 Mar 2023

We wish to thank both Editor and Reviewers for their helpful comments that helped improving our manuscript.

Editor’s comment You indicated that ethical approval was not necessary for your study. We understand that the framework for ethical oversight requirements for studies of this type may differ depending on the setting and we would appreciate some further clarification regarding your research. Could you please provide further details on why your study is exempt from the need for approval and confirmation from your institutional review board or research ethics committee (e.g., in the form of a letter or email correspondence) that ethics review was not necessary for this study? Please include a copy of the correspondence as an ""Other"" file.

Thank you for raising this point.

French law on research involving human subjects applies only to research involving patients and to surveys involving health data. It therefore does not apply to an anonymized survey of medical student feeling about their medical school policy. No personal data was involved. In a study like ours, no IRB approval is required by French law. Obviously students were informed of the purpose of the survey and were perfectly free to refuse it (see response to Reviewer 1). We have clarified the issue in the revised version (page 6). 

Responses to reviewers’ comments

Reviewer #1: Thank you for the opportunity to review this paper. I enjoyed reading it and it reports on a useful and important study. I have made some suggestions for how its clarity could be improved. Most importantly, there is confusion about what the study actually entailed.

Thank you very much for your positive appreciation

As an initial comment, Charte is Charter (rather than chart)

We wrongly thought that both “chart” and “charter” could be used as translation of the French word “charte” Thank you for your comment. The word “chart” has been replaced by “charter” throughout our manuscript. 

In the introduction you say there is a quasi-complete lack of academic policy, but in the next sentence you introduce the Charter. This seems contradictory. Do you mean that there is policy that is not implemented? Or that despite agreeing to the Charter no policy actions have been taken? This needs a bit of clarity.

Thank you for allowing us to clarify this point: there was a lack of policy action or a lack of their evaluation (probably both) in French medical schools, including ours. The purpose of our study is to describe the issue in our medical school. We have modified this sentence page 3 .

The methods section is confusing. Were items 1-4 part of the study or would they have happened anyway as part of professional training? Here they seem to be presented as part of the experiment but up to this point the paper doesn’t suggest this. My reading of the paper is that the survey of the students is the study, to see whether training their instructors on the Charter led to change in practice or had any sort of trickle down effect to students. If this is the case, the info from 1-4 could go into the introduction. I agree that it is important information but it is not a study method. If the activities in 1-4 were done for the specific purpose of this study then that needs to be made much clearer up front and ideally methodologically justified.

The reviewer is right. The text has been modified and point 1-4 no longer appear in the Methods section but in the introduction. 

Would you have expected that working with instructors would sensitise students to COI? Were instructors asked to train students on COI?

The Reviewer raises an important point. One can imagine that working with instructors (this was not the case in our study) would sensitise students. According to their question, we now mention at the end of the paper that our next survey should investigate how students are sensitised to COI by the mandatory course 

Per number 5, was it compulsory? The response rate in the results suggests not but the wording in 5 suggests yes. Perhaps reword 5 in the methods.

This is an important point. Responses were anonymized and students were free to answer or not as mentioned in revised manuscript page 5 

Per number 6, it’s unclear why there are 4 responses to the questions because they are all essentially y/n questions and you subsequently report the results as y/n in each category. Can you talk about why you set up the questionnaire like this?

How did you decide the questions? Are the questions relevant to the Charter? Do they test particular aspects of the Charter?

We explain our choice in the revised manuscript page 5: The questions took into account every point of the charter summary. One to 4 scale was chosen in order to allow nuanced analysis.

Discussion:

The second paragraph repeats the information in the introduction and should be removed. In this paragraph you use ref 8 to note that students are unaware of the harms that COI might pose. This would be useful info to have in the introduction (where you use ref 8 to make a very different point).

Thank you for this pertinent suggestion. We have moved this paragraph from the discussion to the introduction in order to mention the two points addressed in ref 8

In para 3 of the discussion you need to include more information about FORMINDEP if you’re going to talk about survey response. Perhaps at the beginning of this para you could say that the survey measures schools’ performance on COI policy. As it was I wasn’t sure what the numbers referred to and it needed some reading between the lines.

Thank you for allowing us to clarify this issue. We have modified the text according to your recommendation (page 8)

P8 “The lack of knowledge of these issues…” – which issues? Is it that they lack knowledge about COI? About the importance of policies mitigating against COI? or don’t care or haven’t made it a priority or something else?

Also p8 you make it sound like items 1-4 in the methods were in fact part of the study? Again, confusing.

Thank you very much for this comment. We have clarified this confusing point and modified the entire paragraph (page 8) 

P8 is the first place that you mention Charter expectations (eg that teachers should declare at the beginning of each class), though you say you alluded to it. I note in the text that you included the Charter as an appendix but it did not show up in my review. If it is short enough it would be great to include it (or a summary) in a box in the manuscript. It is unlikely that most readers will take the trouble to find an appendix before reading the paper and the paper makes less sense without knowing the contents or even the gist of the Charter.

We are sorry that you could not access the Charter. We have made sure that the summary of the charter is uploaded in appendix

The pitfall section: The sentence about not taking the potential influence of partnership between teachers and industry into account is confusing. Into account how? Perhaps you mean that you only studied the Charter and its implementation and not COI that are not covered in the Charter? Some clarity here would be beneficial.

The Reviewer is right. We have clarified this point by mentioning that our study focused only on students’ perception of the charter implementation

Reviewer #2: Thank you for conducting this study, the results of which seem very important to me. I just have a few minor suggestions to further improve the quality of the paper. I will make my recommendations by mentioning systematically and in order the number of the page to which I refer.

Thank you very much for your positive appreciation.

Page 3:

"This association aims at both to promotion of evidence-based medicine and prevention of the influence of economic interests of industry on patients, doctors, medical students, and public decision-makers" - The sentence does not sound good. It could be something like : « This association aims both to promote evidence-based medicine and to prevent the influence of industry's economic interests on patients, physicians, medical students and public policy makers. »

Thank you for suggesting a better formulation that is now used in our manuscript.

Page 4:

You refer to an appendix but I have not been able to find it

We are sorry that you could not access the Charter. We have made sure that the summary of the charter is uploaded in appendix

Methods, point 3: "Implementation of the Chart" - It is strange to have a specific title for this point and not for the others, it should be harmonized

The Reviewer is right. We have removed this title 

Page 5:

Methods: In my opinion, only points 5 and 6 really constitute the methodology of the survey presented in the study. The previous points seem to me to be more contextualizing elements. Also, it would be interesting to know a bit more about the creation of the questionnaire: how the 10 items were decided, when exactly the questionnaire was submitted, how did you analyse the results, etc.

Both Reviewers rightly make the same point. Point 1-4 are no longer present in the study method and appear in the introduction

Page 7:

Discussion: "the first taking directly in account students’" - "I would add "on this specific subject" or "in France" because there are many international studies that directly take into account the opinions of medical students on their interactions with industry, the issue of independence and their perceptions. For example: https://doi.org/10.1186/s12909-018-1394-9"

The Reviewer is perfectly right: the sentence has been modified accordingly and the reference to the paper of Saito and al added (page 8). 

Page 9:

First, the rather good response rate shows the feasibility of this survey. - « of this kind of survey » ?

We have made this change

-----

As you can see, my main point is about the presentation of the survey methodology. This section could be reworked in order to better present the different steps that led to the creation of the study.

Thank you for pointing this. We have modified the text according to your suggestion (pages 4-5)

For the rest, it is a very good paper on a fundamentally important topic.

Once again, we thank the Reviewer for their positive appreciations

---

## [Decision Letter · Decision Letter 1]

11 Apr 2023

PONE-D-22-34803R1Toward improvement of knowledge of financial conflicts of interest

in a large medical school in FrancePLOS ONE

Dear Dr. alexandra,

Thank you for submitting your manuscript to PLOS ONE. Before a final decision can be made, we invite you to submit a revised version of the manuscript that addresses the remaining points raised by reviewer 2.

We look forward to receiving your revised manuscript.

Kind regards,

Alberto Molina Pérez, Ph.D.

Academic Editor

PLOS ONE

Journal Requirements:

Reviewers' comments:

Reviewer's Responses to Questions

**Comments to the Author**

1. If the authors have adequately addressed your comments raised in a previous round of review and you feel that this manuscript is now acceptable for publication, you may indicate that here to bypass the “Comments to the Author” section, enter your conflict of interest statement in the “Confidential to Editor” section, and submit your "Accept" recommendation.

Reviewer #1: All comments have been addressed

Reviewer #2: All comments have been addressed

2. Is the manuscript technically sound, and do the data support the conclusions?

Reviewer #1: (No Response)

Reviewer #2: Yes

3. Has the statistical analysis been performed appropriately and rigorously? 

Reviewer #1: (No Response)

Reviewer #2: Yes

4. Have the authors made all data underlying the findings in their manuscript fully available?

Reviewer #1: (No Response)

Reviewer #2: Yes

5. Is the manuscript presented in an intelligible fashion and written in standard English?

Reviewer #1: (No Response)

Reviewer #2: Yes

6. Review Comments to the Author

Reviewer #1: (No Response)

Reviewer #2: Thank you for this revised version and for incorporating the comments made earlier. I still have a few minor comments.

Throughout the manuscript, the use of the term "charter" instead of "chart" should be harmonized: there are still quite a few passages where the word "chart" is used.

Page 7: "This questionnaire was also sent to the Conference of the Deans who did not answer it..." -: I think that the use of ellipsis is not very common in scientific articles.

Page 7-8: In the paragraph "The accuracy of results of previous studies is beyond doubt... his amendment occurred after our survey was conducted." - to me, here there are two different ideas: the limitations of the Formindep study and the tools put in place to limit COI. The paragraph should be split in two in order to maintain coherence.

Page 9: "Last, we did not investigate the behavior of the teachers, but they probably share responsibility for the poor results with the students." - I don't understand what you mean here. From the turn of phrase, it is understood that the students would share some responsibility for the poor results of the study. I think that needs to be clarified because the responsibility for the low knowledge of the charter does not lie with them at all.

Thank you for your attention to my comments and congratulations on the great work done on this article.

7. PLOS authors have the option to publish the peer review history of their article (what does this mean?). If published, this will include your full peer review and any attached files.

Reviewer #1: **Yes: **Jane Williams

Reviewer #2: **Yes: **Lucas Bechoux

---

## [Author Response · Author response to Decision Letter 1]

17 Apr 2023

We thank the Reviewer for his positive appreciation of our work and for his helpful suggestions that have all been followed in the manuscript.

Throughout the manuscript, the use of the term "charter" instead of "chart" should be harmonized: there are still quite a few passages where the word "chart" is used.

Done

Page 7: "This questionnaire was also sent to the Conference of the Deans who did not answer it..." -: I think that the use of ellipsis is not very common in scientific articles.

We agree. We have suppressed the suspension points and slightly changed the sentence. 

Page 7-8: In the paragraph "The accuracy of results of previous studies is beyond doubt... his amendment occurred after our survey was conducted." - to me, here there are two different ideas: the limitations of the Formindep study and the tools put in place to limit COI. The paragraph should be split in two in order to maintain coherence.

We have followed the Reviewer’s suggestion and modified the paragraph accordingly. 

Page 9: "Last, we did not investigate the behavior of the teachers, but they probably share responsibility for the poor results with the students." - I don't understand what you mean here. From the turn of phrase, it is understood that the students would share some responsibility for the poor results of the study. I think that needs to be clarified because the responsibility for the low knowledge of the charter does not lie with them at all.

The Reviewer is perfectly right. We have modified the unfortunate sentence.

Thank you for your attention to my comments and congratulations on the great work done on this article

Many thanks again for your very insightful review.

---

## [Editor Report · Decision Letter 2]

4 May 2023

Toward improvement of knowledge of financial conflicts of interest

in a large medical school in France

PONE-D-22-34803R2

Dear Dr. alexandra,

We’re pleased to inform you that your manuscript has been judged scientifically suitable for publication and will be formally accepted for publication once it meets all outstanding technical requirements.

Kind regards,

Alberto Molina Pérez, Ph.D.

Academic Editor

PLOS ONE
---

## [Editor Report · Acceptance letter]

9 May 2023

PONE-D-22-34803R2 

Toward improvement of knowledge of financial conflicts of interest in a large medical school in France

Dear Dr. Alexandra:

I'm pleased to inform you that your manuscript has been deemed suitable for publication in PLOS ONE. Congratulations! Your manuscript is now with our production department. 

Kind regards, 

on behalf of

Dr. Alberto Molina Pérez 

Academic Editor

PLOS ONE